

# Role of methyltransferase-like enzyme 3 and methyltransferase-like enzyme 14 in urological cancers

Zijia Tao, Yiqiao Zhao and Xiaonan Chen

Department of Urology, Shengjing Hospital of China Medical University, Shenyang, Liaoning, China

## ABSTRACT

N6-methyladenosine (m6A) modifications can be found in eukaryotic messenger RNA (mRNA), long non-coding RNA (lncRNA), and microRNA (miRNA). Several studies have demonstrated a close relationship between m6A modifications and cancer cells. Methyltransferase-like enzyme 3 (METTL3) and methyltransferase-like enzyme 14 (METTL14) are two major enzymes involved in m6A modifications that play vital roles in various cancers. However, the roles and regulatory mechanisms of METTL3 and METTL14 in urological cancers are largely unknown. In this review, we summarize the current research results for METTL3 and METTL14 and identify potential pathways involving these enzymes in kidney, bladder, prostate, and testicular cancer. We found that METTL3 and METTL14 have different expression patterns in four types of urological cancers. METTL3 is highly expressed in bladder and prostate cancer and plays an oncogenic role on cancer cells; however, its expression and role are opposite in kidney cancer. METTL14 is expressed at low levels in kidney and bladder cancer, where it has a tumor suppressive role. Low METTL3 or METTL14 expression in cancer cells negatively regulates cell growth-related pathways (e.g., mTOR, EMT, and P2XR6) but positively regulates cell death-related pathways (e.g., P53, PTEN, and Notch1). When METTL3 is highly expressed, it positively regulates the NF-kB and SHH-GL1pathways but negatively regulates PTEN. These results suggest that although METTL3 and METTL14 have different expression levels and regulatory mechanisms in urological cancers, they control cancer cell fate via cell growth- and cell death-related pathways. These findings suggest that m6A modification may be a potential new therapeutic target in urological cancer.

## INTRODUCTION

Chemical modifications of eukaryotic RNA have been known for decades. However, the roles of these modifications in tumor development were largely unknown until recent years. According to the data of MODOMICS, a database of RNA modification pathways, 163 different RNA chemical modifications have been identified in all living organisms (*Boccaletto et al., 2018*). The N6-methyladenosine (m6A) modification is one of the most common, invertible, and abundant modifications found on eukaryotic mRNA, miRNA, lncRNAs, and other RNA molecules. These modifications affect the

Corresponding author
Xiaonan Chen, chenxn@cmu.edu.cn

transcription, processing, translation, and metabolism of these RNA molecules (*Zheng et al., 2019*). The m6A modification occurs by a dynamic process involving three major classes of enzymes: 'Writers,' 'Erasers,' and 'Readers' (*Vu, Cheng & Kharas, 2019*). Writers include methyltransferase-like enzyme 3 (METTL3), methyltransferase-like enzyme (METTL14), Wilms tumor 1-associated protein (WTAP), RNA binding motif protein 15/15B (RBM15/15B), and vir-like M6A methyltransferase-associated (VIRMA), which catalyze the generation of m6A. Erasers, which include fat and obesity-related protein (FTO) and alkB homolog 5 (ALKBH5), are responsible for demethylation. Readers recognize the m6A methylation and generate functional signals (*Chen, Zhang & Zhu, 2019b*). This latter class of enzymes includes eukaryotic initiation factor (eIF3), the IGF2 mRNA binding protein (IGF2BP) family, the heterogeneous nuclear ribonucleoproteins (HNRNP) protein family, and proteins that contain a YT521-B homology (YTH) domain.

METTL3 is a 70-kDa protein that was first identified in Hela cell lysates (*Bokar et al., 1997*). It contains two domains that bind S-adenosylmethionine (SAM) and catalyze the formation of m6A in RNA (*Leach & Tuck, 2001*). WTAP promotes METTL3 localization to nuclear spots and greatly improves its catalytic activity (*Ping et al., 2014*). Studies have also shown that METTL3 acts as a positive regulator of mRNA translation independent of methyltransferase activity: promoting translation by involving in translation initiation mechanisms in the cytoplasm (*Ke et al., 2017*). It has been reported that METTL3 can play the role without METTL14 and can promote translation of specific mRNAs independently of its catalytic activity in vitro (*Ke et al., 2017*). METTL3 is the core catalytic activity in the N6-methyltransferase complex formed by the METTL3-METTL14 heterodimer. Adenosine residues at the N (6) position of some RNAs are methylated by this complex (*Alarcon et al., 2015a*; *Alarcon et al., 2015b*; *Bokar et al., 1997*; *Dominissini et al., 2012*; *Du et al., 2018*; *Liu et al., 2015*; *Meyer et al., 2015*; *Scholler et al., 2018*; *Sledz & Jinek, 2016*; *Wang, Doxtader & Nam, 2016a*; *Wang et al., 2016b*; *Wang et al., 2014*; *Xiang et al., 2017*; *Zhong et al., 2018*). METTL14 is a scaffold for bound RNA and identifies the substrate of the N6-methyltransferase complex formed by the METTL3-METTL14 heterodimer (*Liu et al., 2014*; *Liu et al., 2015*; *Ping et al., 2014*; *Scholler et al., 2018*; *Sledz & Jinek, 2016*; *Wang, Doxtader & Nam, 2016a*; *Wang et al., 2016b*). METTL14 shares about 22% sequence identity and nearly identical topology with domains found in METTL3. When part of the METTL3-METTL14 heterodimer, METTL14 is thought to assume a pseudo-methyltransferase function that helps bind RNA and stabilize METTL3. However, it is possible that methyltransferase activity mediated by METTL14 may occur after the binding of additional factors (*Wang et al., 2017*). In mRNA, the methylation site is located in the 5′-[AG] GAC-3′consensus site found in some mRNAs, which plays an important role in mRNA stability, processing, translation efficiency, and editing (*Alarcon et al., 2015a*; *Alarcon et al., 2015b*; *Bokar et al., 1997*; *Dominissini et al., 2012*; *Liu et al., 2015*; *Meyer et al., 2015*; *Wang et al., 2014*; *Xiang et al., 2017*). Methylation is completed after the mRNA is released into the nucleoplasm and promotes mRNA instability and degradation (*Ke et al., 2017*).

In recent years, the role of m6A in various cancers, including leukemia, brain, cervical, endometrial, breast, liver, and lung cancer, has been revealed (*Chen et al., 2018*; *Choe*

*et al., 2018*; *Liu et al., 2018*; *Vu et al., 2017*; *Weng et al., 2018*; *Zhang et al., 2016*; *Zhang et al., 2017*). m6A serves a regulatory function in oncogenesis and development by modifying many target genes (*Deng et al., 2018*; *Liu et al., 2018*). Interestingly, m6A may have oncogenic or suppressive functions depending on the cellular environment (*Cui et al., 2017*; *Li et al., 2017b*; *Lin et al., 2016*; *Ma et al., 2017*; *Visvanathan et al., 2018*; *Vu et al., 2017*; *Zhang et al., 2016*; *Zhang et al., 2017*). METTL3 mediates YTHDF2-dependent post-transcriptional silencing of SOCS2(Suppressor of Cytokine Signaling 2) to promote liver cancer progression (*Chen et al., 2018*). METTL14 is expressed at low levels in liver cancer and hematopoietic stem cells, and it impairs acute myelocytic leukemia (AML) oncogenesis (*Weng et al., 2018*). METTL14 can also inhibit liver oncogenesis and metastasis (*Li et al., 2017b*). Some studies have shown that abnormal m6A modification is necessary for oncogenesis and progression (*Cui et al., 2017*; *Li et al., 2017b*; *Lin et al., 2016*; *Ma et al., 2017*; *Visvanathan et al., 2018*; *Vu et al., 2017*; *Zhang et al., 2016*; *Zhang et al., 2017*), suggesting that the pathway involved in the m6A modification may be a promising therapeutic target in oncology.

Urological tumors include kidney, bladder, prostate, and testicular cancer. After decades of research, there have been significant improvements in the treatment of these cancer types; however, drug resistance and low survival rates still prevail. In addition, the lack of accurate and useful molecular markers for timely diagnosis and prognosis assessment of patients has led to unsatisfactory treatment results (*Cai et al., 2019*; *Cheng et al., 2018*; *Cheng et al., 2019*; *Gong et al., 2019*). In this review, we summarize the present research progress in understanding the roles of METTL3 and METTL14 in urological tumors and their potential as treatment and diagnostic markers.

## SURVEY METHODOLOGY

In order to search literatures exhaustively, we used keywords 'METTL3', 'Methyltransferase-like enzyme 3', 'METTL14', 'Methyltransferase-like enzyme 14', 'kidney cancer', 'renal cell carcinoma', 'bladder cancer', 'prostate cancer', and 'testicular cancer' to search articles in the PubMed, Web of Science and CNKI. We excluded the articles which were not associated with METTL3, METTL14 and urological cancers.

## METHYLTRANSFERASE IN UROLOGICAL CANCERS

### Kidney cancer

According to the GLOBOCAN (Global Cancer Observatory) statistics, 403,262 people were diagnosed with renal cancer throughout the world, and 175,098 people died in 2018, making renal cancer the 14th most common cancer in the world. There is a higher incidence of this cancer in males than in females (*Bray et al., 2018*). Many patients remain asymptomatic until renal masses progress to an advanced stage because of the position of the kidneys in the body. Based on the World Health Organization (WHO) 2016 classification, renal cell carcinoma(RCC) are divided into three main subcategories: (1) clear cell renal cell carcinoma (ccRCC), the most common and aggressive type; (2) chromophobe renal cell carcinoma (chRCC); (3) papillary renal cell carcinoma (pRCC), which consists of types 1

and 2 (*Hao et al., 2019*; *Moch et al., 2016*). Analysis of methyltransferases in kidney cancer, including ccRCC, indicates that both METTL3 and METTL14 are tumor suppressors in this disease (*Gong et al., 2019*; *Li et al., 2017a*; *Wang et al., 2019*; *Zhou et al., 2019*).

### METTL3 in kidney cancer

METTL3 is more prone to copy number variations (CNV) or mutation than other genes in ccRCC, and patients affected by METTL3 shallow deletions (a form of CNV) have poorer disease-free survival (DFS) and overall survival (OS) (*Zhou et al., 2019*). METTL3 mRNA and protein expression are low in RCC. Its expression level is negatively related to higher histological grade, larger tumor size, shorter OS, and shorter DFS (*Li et al., 2017a*; *Zhou et al., 2019*). Knocking down of METTL3 expression in RCC cell lines significantly increases proliferation, migration, and invasion (*Li et al., 2017a*). The VHL-HIF-ZNF217(Von Hippel-Lindau- Hypoxia Inducible Factor- Zinc Finger Protein 217)-METTL3 pathway may be involved in m6A regulation in ccRCC cells by mediating two downstream m6A targets, the PI3K/AKT/mTOR (Phosphatidylinositol 3-Kinase/AKT, also known as Protein Kinase B/mammalian Target of Rapamycin) and p53 signaling pathways (*Li et al., 2017a*; *Zhou et al., 2019*). The PI3K/AKT/mTOR pathway plays a significant role in cell proliferation, growth, and survival (*O'Reilly et al., 2006*; *Shaw & Cantley, 2006*). In addition, METTL3 may inhibit the invasion and migration of RCC through the epithelial-mesenchymal transition (EMT) pathway (*Li et al., 2017a*).

GSEA analysis of ccRCC patient tumors suggests that low METTL3 expression levels may be related to some critical biological processes, such as the mTOR pathway, adipogenesis, and reactive oxygen species (ROS), which partially validates the RCC cell line results. Based on these data, the mTOR pathway may be the key target of the m6A modification in kidney cancer. Alternatively, METTL3 also regulates the cell cycle. Downregulation of METTL3 significantly decreases G1 cell cycle arrest, whereas the upregulation of METTL3 increases G1 arrest (*Zhou et al., 2019*).

### METTL14 in kidney cancer

Like METTL3, patients with kidney cancer are more predisposed to METTL14 mutations or CNV, and patients affected by shallow deletions of METTL14 have a poorer OS and DFS (*Zhou et al., 2019*). METTL14 is mainly located in the nucleus of ccRCC cells. Compared with normal kidney tissues, METTL14 mRNA expression is significantly lower in ccRCC tumors. METTL14 expression levels are negatively correlated with RCC pathological and clinical stages, and positively correlated with OS (*Wang et al., 2019*). As with METTL3, the VHL-HIF-ZNF217-METTL14 pathway regulates m6A in ccRCC cells via the PI3K/AKT/mTOR and p53 signaling pathways (*Li et al., 2017a*; *Zhou et al., 2019*). METTL14 has also been associated with two other regulating pathways, including P2RX6(Purinergic Receptor P2X 6) and PTEN (Phosphatase and Tensin Homolog) (*Gong et al., 2019*; *Wang et al., 2019*). P2RX6 is a non-selective cation channel protein that is a preferred receptor for ATP (*Chadet et al., 2014*; *North, 2002*). METTL14 expression is negatively correlated with P2RX6. Low METTL14 expression is associated with a shorter OS, while low P2RX6 expression correlates with a longer OS. METTL14 may increase the pre-mRNA splicing of P2RX6 by increasing the methylation of P2RX6 mRNA, thereby

inhibiting P2RX6. Low METTL14 expression in cancer cells leads to high P2RX6 expression via the ATP-P2RX6-Ca2+-p-ERK1/2(Extracellular Regulated protein Kinases 1/2)-MMP9 (Matrix Metallopeptidase 9) signaling pathway, which promotes renal tumor cell metastasis and invasion (*Gong et al., 2019*). PTEN is a tumor suppressor, whose duty is to encode phosphatidylinositol-3,4,5-trisphosphate 3-phosphatase to preferentially dephosphorylate phosphoinositide substrates. The METTL14 mRNA expression level is positively associated with PTEN. Patients with low METTL14 and PTEN expression levels have a shorter OS. METTL14 stabilizes PTEN mRNA by regulating the m6A levels on the PTEN mRNA. PTEN acts as a tumor suppressor by negatively regulating the AKT/PKB signaling pathway. Synergistic effects may occur through the interaction of EIF3A (Eukaryotic Translation Initiation Factor 3 Subunit A) and METTL14, which regulates kidney cancer progression. In addition, 24 circRNAs (e.g., circ-0023414 and circ-0031772) interact with four miRNAs (miR-130a-3p, miR-106b-5p, miR-130b-3p, and miR-301a-3p), which have a negative relationship with METTL14 mRNA (*Wang et al., 2019*). These circRNAs may act as miRNA sponges to regulate METTL14 mRNA (Fig. 1).

## Bladder cancer

Bladder cancer was the 12th most common cancer globally in 2018, with 549,393 newly diagnosed cases and 199,922 deaths. The incidence of bladder cancer varies by gender, with males at higher risk (*Bray et al., 2018*). Urothelial carcinoma is a common histological type of bladder cancer. Non-papillary muscle-invasive and papillary non-muscle-invasive tumors are the two main types of this disease (*Sanli et al., 2017*; *Wu et al., 2019*). Most of the available research on methyltransferases in bladder cancer suggests that METTL3 is a oncogene, whereas METTL14 is a tumor suppressor.

### *METTL3 in bladder cancer*

METTL3 is highly expressed in bladder cancer tissues (*Chen et al., 2019a.*). Overexpression of METTL3 significantly promotes the growth and invasion of bladder tumor cells. In contrast, METTL3 knockdown abrogates the proliferation, invasion, and viability of bladder cancer cells and reduces the proportion of cells in the S phase of the cell cycle while increasing the proportion in G1. METTL3 may maintain the characteristics of bladder cancer stem cells by inducing the m6A modification of SOX2(SRY-Box Transcription Factor 2), a marker of bladder cancer stem cells both in vivo and in vitro (*Zhu et al., 2017*). Patients with high METTL3 expression in bladder cancer have higher histological scores, worse prognosis, and shorter survival time. Thus, METTL3 exhibits a oncogenic role in bladder cancer.

The AFF4/NF-$\kappa$B/MYC (AF4/FMR2 Family Member 4/ Nuclear Factor Kappa B/ Myelocytomatosis oncogene) signaling network plays a vital role in the upregulation of METTL3 in bladder cancer. METTL3 can directly increase the abundance of m6A sites on the MYC mRNA to improve the stability of MYC transcripts and increase MYC protein expression. It has a similar effect on AFF4 mRNA and protein. AFF4 protein directly binds to the MYC promoter to extend MYC transcription and upregulate MYC expression. METTL3 may also promote the expression of IKBKB (Inhibitor of Nuclear Factor Kappa B

**METTL3/14 regulatory network in kidney cancer.**

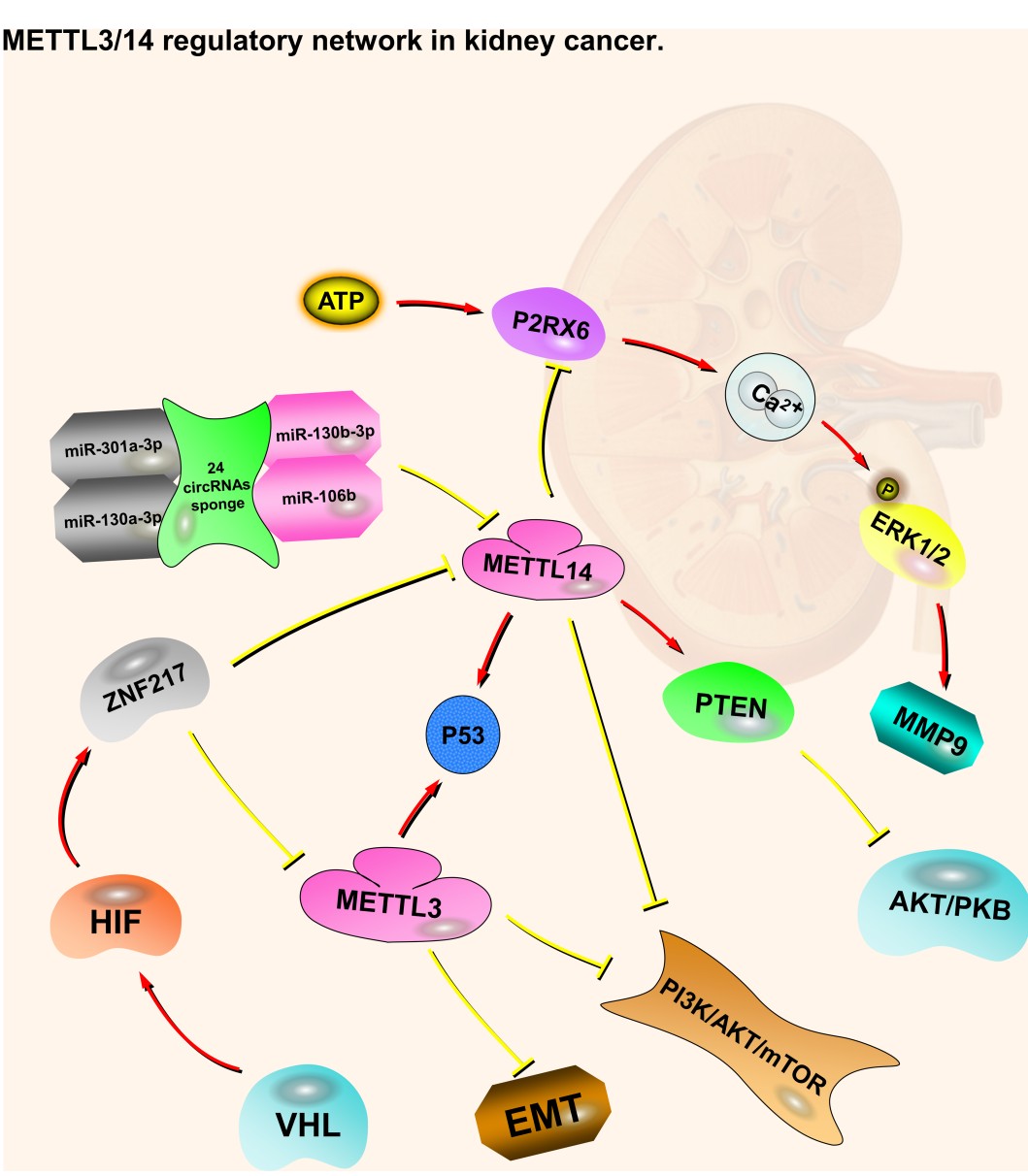

**Figure 1** **METTL3 and METTL14 regulatory network in kidney cancer.** The red arrows in the figure represent the promoting effect, and the yellow arrows represent the inhibiting effect. PI3K/AKT/mTOR, EMT and P2RX6 play an oncogenic role in kidney cancer, while p53 and PTEN play a tumor suppressive role. METTL3 and METTL14 play a role in suppressing kidney cancer by inhibiting or promoting some pathways, respectively. At the same time, they also accept regulation from upstream molecules or pathways.

Kinase Subunit Beta) and RELA (V-Rel Reticuloendotheliosis Viral Oncogene Homolog A), which are two key regulators of the NF-$\kappa$B pathway, by regulating translation efficiency and subsequently inducing MYC transcription. Thus, m6A modifications mediated by METTL3 through different signaling pathways converge at MYC expression. This m6A-regulated malignant regulatory network effectively increases MYC protein levels in bladder cancer

and may lead to difficulties in reducing MYC by blocking a single signaling pathway (*Cheng et al., 2019*).

The METTL3-DGCR8 (DiGeorge Syndrome Critical Region 8)-pri-mi221/222-PTEN pathway also mediates the upregulation of METTL3 in bladder cancer. METTL3 can actively regulate pri-miR221/222 in an m6A-dependent manner by interacting with DGCR8, a micro-processor protein that promotes the processing of pri-miR221/222 into mature miR221/222 in bladder cancer. miR221/222 binds to the 3′-untranslated region (UTR) of PTEN mRNA, leading to decreased PTEN mRNA and protein expression (*Han et al., 2019*).

Three other pathways are also associated with METTL3 upregulation in bladder cancer, including the METTL3 -CDCP1 (CUB Domain Containing Protein 1), METTL3-ITGA6(Integrin Subunit Alpha 6), and METTL3/YTHDF2-SETD7/KLF4(SET Domain Containing Lysine Methyltransferase 7/ Kruppel Like Factor 4) m6A axes. METTL3 and CDCP1 are both upregulated in bladder cancer patient samples and related to bladder cancer progression. Inhibition of the METTL3-CDCP1 axis reduces the growth and progression of bladder cancer cells and chemical-transformed cells. The METTL3-CDCP1 axis and chemical carcinogens have synergistic effects on the malignant transformation of uroepithelial cells and bladder cancer oncogenesis (*Yang et al., 2019*). In the METTL3-ITGA6 axis, METTL3 highly enriches the m6A methylation levels in the ITGA6 mRNA 3′-UTR region, which promotes the translation of ITGA6 mRNA. The binding of YTHDF1/YTHDF3 to the m6A motif in the ITGA6 3′-UTR region further increases ITGA6 translation. This overexpression of ITGA6 increases the adhesion, proliferation, and migration of bladder tumor cells and enhances their metastasis. Therefore, ITGA6 is a crucial target of METTL3 function in bladder cancer (*Jin et al., 2019*). METTL3 also catalyzes m6A modifications in the mRNAs of SETD7 and KLF4, two tumor suppressors that are part of the METTL3/YTHDF2-SETD7/KLF4 m6A axis. YTHDF2 recognizes these m6A modifications and degrades the SETD7 and KLF4 mRNAs, leading to bladder cancer progression (*Xie et al., 2020*).

Although most studies suggest that METTL3 can foster bladder cancer growth and progression, one study suggests that METTL3 can act as a bladder tumor suppressor. *Zhao et al. (2019)* identified METTL3 as a driver gene in a bladder cancer cohort using the integrated statistical model-based method called driver MAPS. However, in the subsequent experimental verification of this finding, the researchers found that METTL3 knockdown significantly increased cell proliferation. Furthermore, METTL3 somatic mutations could promote cancer cell growth by interrupting RNA methylation. Therefore, they believe that METTL3 acts as a tumor suppressor for bladder cancer (*Zhao et al., 2019*).

### METTL14 in bladder cancer

METTL14 is expressed at low levels in bladder cancer and bladder tumor-initiating cells (TICs). Bladder cancer TICs possess self-renewal, differentiation, and tumor-initiating properties. METTL14 inhibits these properties along with the maintenance and metastasis of bladder TICs. METTL14 expression is negatively associated with the severity of bladder

cancer and clinical outcome. METTL14 is significantly related to the T stage of the TNM stage system (*Chen et al., 2019a.*).

Notch1 plays an important part in bladder oncogenesis and TICs self-renewing. It is a downstream target of METTL14. m6A modification of Notch1 (Neurogenic Locus Notch Homolog Protein 1) decreases its RNA stability, leading to inhibition of bladder cancer and bladder tumor-initiating cells (*Gu et al., 2019*). Thus, METTL14 is a tumor suppressor in bladder cancer, acting through the METTL14-Notch1 pathway (Fig. 2).

## Prostate cancer

Prostate cancer was the third most common cancer worldwide in 2018, with 1,276,106 newly diagnosed cases and 358,989 deaths (*Bray et al., 2018*). Because of the aging of the growing population, prostate cancer has become a main public health problem for men (*Center et al., 2012*). This tumor is often silent in clinical practice and usually found after it invades other tissues (*Guo et al., 2019*; *Roobol & Carlsson, 2013*; *Shen & Abate-Shen, 2010*). Studies of METTL3 in prostate cancer suggest that it is a oncogene.

METTL3 protein and mRNA expression levels in prostate cancer are significantly higher than those in adjacent benign tissue. METTL3 is mainly localized to the nucleus of prostate cells, with a small amount in the cytoplasm. METTL3 mRNA and protein levels are positively correlated with prostate-specific antigen (PSA) values and Gleason scores. Therefore, METTL3 plays an oncogenic role in prostate cancer and may be used in combination with PSA as a diagnostic marker for this disease (*Xianyong et al., 2019*).

Knockdown of METTL3 in prostate cancer cell lines reduces the m6A content and inhibits survival, cell proliferation, colony formation, and invasion. Mechanistic analysis indicated that there is decreased GLI1(Glioma-Associated Oncogene Family Zinc Finger 1) expression after METTL3 depletion. GLI1 is an important component of the SHH (Sonic Hedgehog Signaling Molecule)-GLI signaling pathway that is positively correlated with prostate cancer severity. GLI1 is a negative modulator of the androgen receptor and contributes to the androgen-independent growth of prostate cancer. c-Myc and cyclin D1 mRNA levels (SHH signaling downstream targets) are also inhibited, resulting in apoptosis (*Cai et al., 2019*).

METTL3 expression is higher in prostate cancer than in normal prostate tissue, especially in prostate cancer with bone metastasis (*Li et al., 2020*). High METTL3 expression is positively correlated with prostate cancer progression and poor prognosis. METTL3 overexpression increases the m6A levels of integrin $\beta$1 (ITGB1) mRNA. HuR (also known as ELAV Like RNA Binding Protein 1) interacts with this modified mRNA to increase its stability and promote protein expression, making prostate cancer cells capable of adhering to collagen I in bone marrow stroma. This finding can explain the mechanism of prostate cancer bone metastasis to some extent. METTL3 also increases the m6A methylation of lymphoid enhancer-binding factor 1 (LEF1) mRNA, which promotes its protein expression and the progression of prostate cancer by activating the Wnt-$\beta$-catenin pathway (*Ma, Cao & Zhao, 2020*). Thus, METTL3 is involved in the regulation of multiple pathways and mechanisms in prostate cancer and may have a pivotal position in this complex regulatory network.

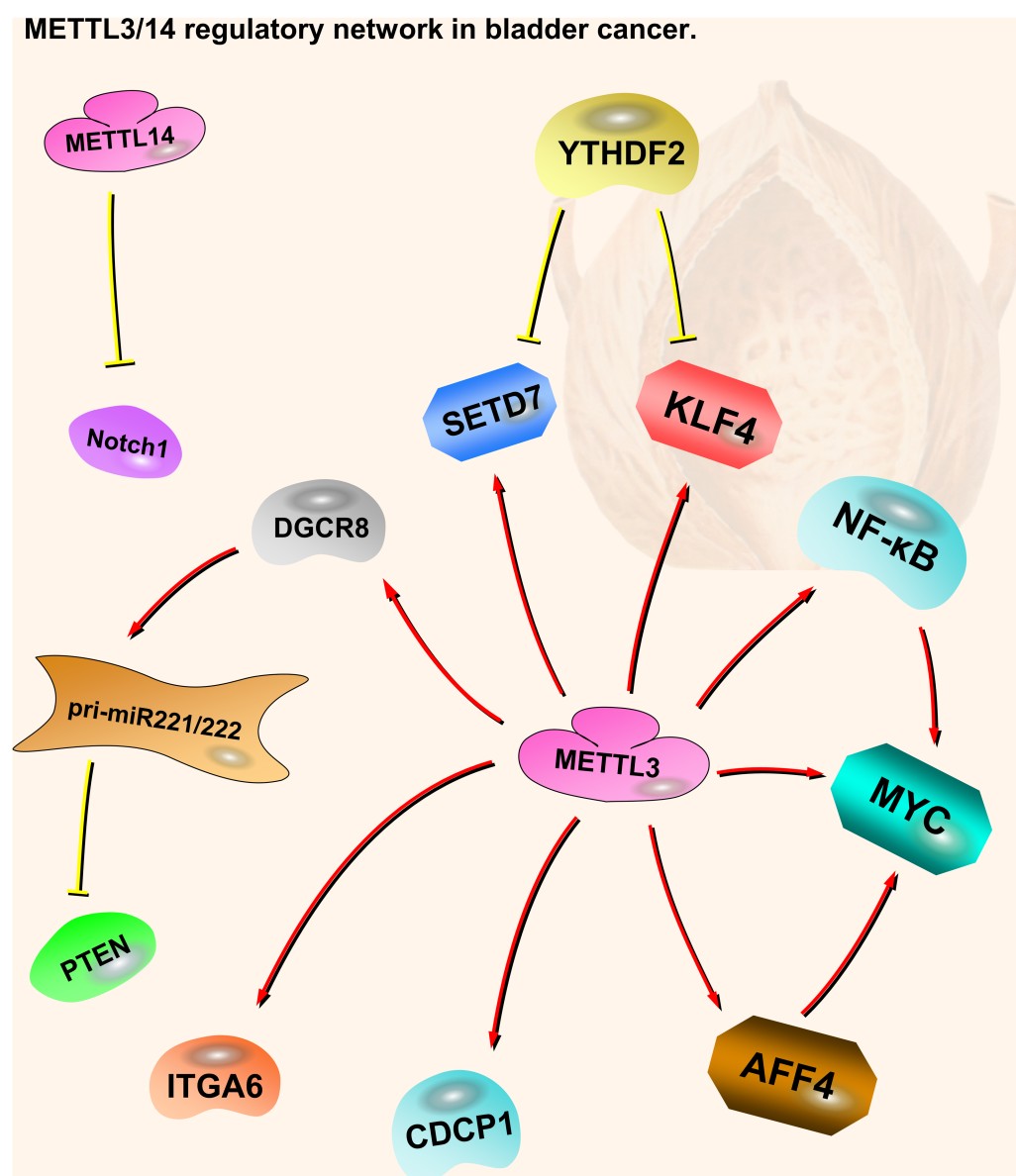

**METTL3/14 regulatory network in bladder cancer.**

**Figure 2** **METTL3 and METTL14 regulatory network in bladder cancer.** The red arrows in the figure represent the promoting effect, and the yellow arrows represent the inhibiting effect. AFF4/NF-$\kappa$ B/MYC, DGCR8-pri-miR221/222-PTEN, CDCP1, ITGA6 and Notch1 play oncogenic role in bladder cancer, while SETD7 and KLF4 play a tumor suppressive role. METTL3 can play an oncogenic role in bladder cancer through a variety of pathways. METTL14 is the opposite; although there are not many related studies, it is certain that it plays a tumor suppressive role in bladder cancer.

Although there are not many reports on m6A in prostate cancer, existing articles describe the mechanisms and research prospects of 'Writer' enzymes. Higher VIRMA expression levels are detected in metastatic castration-resistant prostate cancer (mCRPC) cells. Patients with high VIRMA expression have a significantly shorter disease-free survival. METTL3, METTL14, WTAP, and VIRMA form a methyltransferase complex (MTC);

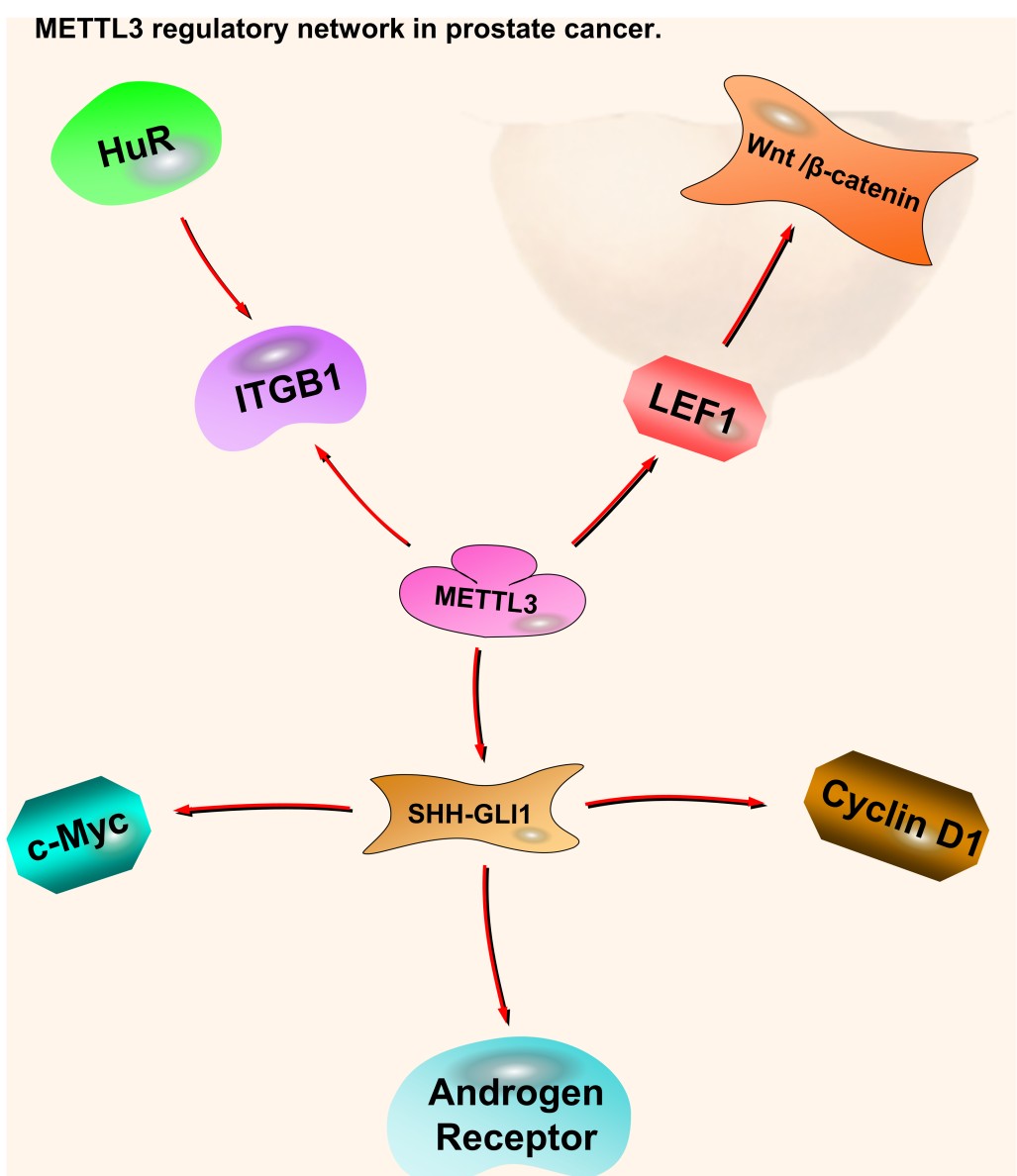

**Figure 3** **METTL3 regulatory network in prostate cancer.** The red arrows in the figure represent the promoting effect. There are not many researches on METTL3 in prostate cancer compared with kidney cancer and bladder cancer. It is certain that it plays a role in oncogenesis by affecting downstream pathway SHH-GLI1, ITGB1, and LEF1-Wnt/$\beta$-catenin. At present, there is no research on the pathway of METTL14 in prostate cancer.

however, each component can function independently in other cellular processes. The knockout of VIRMA triggers a compensatory feedback loop that enhances the expression of the catalytic METTL3 subunit. However, compensatory METTL3 overexpression is insufficient to maintain MTC function without VIRMA (*Barros-Silva et al., 2020*) (Fig. 3).

## Testicular cancer

According to the GLOBOCAN statistics, 71,105 people were diagnosed with testicular cancer globally in 2018, and 9,507 people died (*Bray et al., 2018*). More than 95% of testicular neoplasms are testicular germ cell tumors (TGCTs), which form two subclasses: germ-cell neoplasia in situ (GCNIS)-related and GCNIS-unrelated tumors. The GCNIS-related tumors include seminomas (SEs) and non-seminoma tumors (NSTs) (*Moch et al., 2016*). Although the morbidity and mortality of testicular tumors are not high compared to other urological tumors, there are no accurate and effective biomarkers for treatment (*Lobo et al., 2018*). There have been few studies on METTL3 and METTL14 in testicular cancer. In testicular germ cell tumor cell lines and tissues, METTL3 appears to be the main 'Writer' enzyme. METTL14 expression can be detected but only at moderate levels. Its expression in SEs is significantly higher than in embryonal carcinoma (*Nettersheim et al., 2019*). METTL14 is expressed at lower levels in SEs compared to NSTs (*Lobo et al., 2018*). In contrast, the mRNA expression levels of other m6A-related enzymes (e.g., VIRMA and YTHDF3) in SEs are higher than in NSTs. VIRMA expression is positively correlated with YTHDF3 expression levels. These results suggest that m6A enzymes mainly contribute to the SE phenotype, but not other subtypes. Because of the expression of VIRMA and YTHDF3 in SEs, both enzymes may represent new candidate biomarkers for SE patient management (*Lobo et al., 2019*). So far, we believe that m6A may be a new direction to break through the current dilemma of testicular cancer.

## CONCLUSIONS

From the limited studies available, we found that METTL3 and METTL14 have different expression patterns in four types of urological cancer (kidney, bladder, prostate, and testicular cancer). METTL3 is highly expressed in bladder and prostate cancer, where it plays oncogenic role. In contrast, METTL3 expression is low in kidney cancer. METTL14 is expressed at low levels in kidney and bladder cancer playing tumor suppressive role. High METTL14 expression has not been found in urological cancers. Regardless of the type of urological cancer, low METTL3 or METTL14 expression negatively regulates cell growth-related pathways (e.g., mTOR, EMT, and P2XR6) but positively regulates cell death-related pathways or tumor suppressors (e.g., P53, PTEN, and Notch1). When METTL3 is highly expressed, it positively regulates the NF-kB and SHH-GL1 pathways (proliferation-related pathways) and negatively regulates PTEN (Table 1).

Compared to METTL14, METTL3 seemly shows various expression patterns and affects different regulation pathways depending on the type of urological cancer, suggesting that METTL3 has organ-specific characteristics. Based on available data, modulation of m6A regulation may represent a new therapeutic target for urological cancer treatment. However, because of the limited number of available studies, we cannot fully elucidate the molecular mechanisms regulating the m6A modification in urological tumors. Additional studies are needed to thoroughly understand the mechanism and determine the therapeutic potential of targeting m6A regulation in urological tumors. Based on the existing results, METTL3 and METTL14 control cancer cell fate through cell growth- and cell death-related

Tao et al. (2020), *PeerJ*, DOI 10.7717/peerj.9589

**Table 1**  Summary of METTL3 and METTL14 related pathways in urological tumors.

| Diseases | Component | Role | Source of experimental evidence | Regulation | Potential signal pathway | Author & Refs |
|---|---|---|---|---|---|---|
| | METTL3 | Tumor suppressor | RCC and matched histologically-normal renal tissues are from 145 RCC patients; RCC cell lines (CAKI-1, CAKI-2 and ACHN); a normal renal tubular epithelial cell line (HK-2); BALB/c nude mice | Down-regulation | METTL3-PI3K/AKT/mTOR METTL3-EMT | *Li et al. (2017a)*, *Li et al. (2017b)* |
| | METTL3 | Tumor suppressor | 528 patients with CNV data and pathology reports from the TCGA database; GSEA database | Up-regulation | VHL-HIF-ZNF217-METTL3-PI3K/AKT/mTOR VHL-HIF-ZNF217-METTL3-p53 | *Zhou et al. (2019)* |
| Kidney cancer | METTL14 | Tumor suppressor | 528 patients with CNV data and pathology reports from the TCGA database; GSEA database | Up-regulation | VHL-HIF-ZNF217-METTL14-PI3K/AKT/mTOR VHL-HIF-ZNF217-METTL14-p53 | *Zhou et al. (2019)* |
| | METTL14 | Tumor suppressor | Online databases (TCGAportal,GTExPortal, UALCAN,GEPIA2,MEXPRESS, RMBasev2.0,OncoLnc,starBase, circBank,STRING…) | Down-regulation | circRNAs-miRNAs-METTL14-PTEN-AKT/PKB | *Wang et al. (2019)* |
| | METTL14 | Tumor suppressor | 17 groups of renal cell carcinoma tissues and adjacent tissues received in patients with partial or complete kidney resection; Renal cancer cell line (OS-RC-2,786-O,HEK, -293, SN12-PM6, SW839, A498); Human cortical proximal tubule epithelial cell line (HK-2); Nude mice; Online databases (TCGA, UALCAN…) | Down-regulation | METTL14-P2RX6-Ca2+-p-ERK1/2-MMP9 | *Gong et al. (2019)* |

Tao et al. (2020), *PeerJ*, DOI 10.7717/peerj.9589

**Table 1** (*continued*)

| Diseases | Component | Role | Source of experimental evidence | Regulation | Potential signal pathway | Author & Refs |
|---|---|---|---|---|---|---|
| | METTL3 | Oncogene | Human/mouse bladder cancer samples; bladder cancer cell lines (5637, UM-UC-3); Immortalized epithelial cells (SV-HUC-1); GSEA database | Up-regulation | METTL3-AFF4 /NF- $\kappa$ B/ MYC | *Cheng et al. (2019)* |
| | METTL3 | Oncogene | Human/mouse bladder cancer samples; bladder cancer cell lines (EJ, T24) | Up-regulation | METTL3-DGCR8-pri-miR221/222-PTEN | *Han et al. (2019)* |
| Bladder cancer | METTL3 | Oncogene | Formalin-fixed paraffin-embedded (FFPE) tissue from 114 cases of bladder cancer and 30 cases of cystitis with radical cystectomy and bladder biopsy; Human prostate epithelial cell line (RWPE-1); Human bladder cancer cell line (T24, UM-UC-3); urethral epithelial cells (SV-HUC-1); 3-methylcholesterol transformed urethral epithelial cells (MC-SV-HUC T2);NOD/SCID mice | Up-regulation | METTL3 -CDCP1 | *Yang et al. (2019)* |
| | METTL3 | Oncogene | TCGA GDAC Firehose; bladder cancer cell lines(T24) | Up-regulation | METTL3- ITGA6 | *Jin et al. (2019)* |
| | METTL3 | Oncogene | Human bladder cancer samples (T24, UM-UC-3); normal human urothelium cell line SV-HUC-1 | Up-regulation | METTL3/YTHDF2-SETD7/KLF4 m6A axis | *Xie et al. (2020)* |
| | METTL14 | Tumor suppressor | Primary bladder cancer specimens from 6 bladder cancer patients | Down-regulation | METTL14-Notch1 | *Gu et al. (2019)* |
| Prostate cancer | METTL3 | Oncogene | Human prostate cancer cell lines(LNCaP, PC3, C4-2, C4-2B, DU-145); Human normal prostate epithelial cell line(RWPE-1); Six-week-old male NOD/SCID mice | Up-regulation | METTL3-SHH/GLI1-c-Myc/cyclin D1 | *Cai et al. (2019)* |

Tao et al. (2020), *PeerJ*, DOI 10.7717/peerj.9589

**Table 1** (*continued*)

| Diseases | Component | Role | Source of experimental evidence | Regulation | Potential signal pathway | Author & Refs |
|---|---|---|---|---|---|---|
| | METTL3 | Oncogene | 15 localized prostate cancer tissues with bone metasta­sis and corresponding ad­jacent tissues; the human prostate cancer cells PC3 and LNCaP; twenty 5-week-old male SCID mice | Up-regulation | METTL3/HuR-ITGB1 | *Li et al. (2020)* |
| | METTL3 | Oncogene | 48 prostate cancer tissues and adjacent normal ones (3 cm away from the tumor tis­sues); human prostate cells RWPE-2; human prostate cancer cells PC3 and LNCaP | Up-regulation | METTL3-LEF1-Wnt/$\beta$-catenin | *Ma, Cao & Zhao (2020)* |

pathways. Although METTL3 and 14 have been a prominent focus of studies of m6A in urological tumors, the role of other enzymes is also worth studying. The m6A-related enzymes VIRMA and YTHDF3 have been implicated in testicular cancer. Additional research is needed to define the mechanisms of these m6A enzymes in this disease.

Other researchers have studied the role of m6A-related genes in urological tumors from different directions. Unlike our approach, they mainly focused on analyzing the expression of m6A-related genes using the TCGA database and found that urological cancer tends to follow the same pattern, with the upregulation of methyltransferase related to higher tumor grade and stage. In addition, they looked not only at expression differences of 'Writers' in urological tumors but also differences in the expression of 'Erasers' and 'Readers' in these tumor types. Interestingly, VIRMA is upregulated in all four urological tumors, which indicates that it could be a potential molecular target worth exploring. Although these results are refreshing, the relevant conclusions and opinions must be confirmed experimentally (*Lobo et al., 2018*).

In this review, the potential molecular networks surrounding the m6A modification are described based on existing research. Although m6A has just emerged in urological oncology, it has already shown researchers a promising direction. The available data suggest that regulators of the m6A modification may represent new targets and biomarkers for the treatment and diagnosis or prognosis of urological cancers.

## ACKNOWLEDGEMENTS

Thanks to Professor Han Fang, who serves for PTSD Laboratory of Department of Histology and Embryology in the Basic Medical College of China Medical University, for her strong guidance and support for this review; she has put forward valuable suggestions for the improvement of this paper.

### Funding

This study was supported by the Natural Science Foundation of Liaoning Province of China (Grant No. 20170540988) and Shenyang Science and Technology Program (Grant No. 17-231-1-57). The funders had no role in study design, data collection and analysis, decision to publish, or preparation of the manuscript.

### Grant Disclosures

The following grant information was disclosed by the authors:
Natural Science Foundation of Liaoning Province of China: 20170540988.
Shenyang Science and Technology Program:  17-231-1-57.

### Competing Interests

The authors declare there are no competing interests.

## Author Contributions

- Zijia Tao conceived and designed the experiments, performed the experiments, analyzed the data, authored or reviewed drafts of the paper, and approved the final draft.
- Yiqiao Zhao analyzed the data, prepared figures and/or tables, and approved the final draft.
- Xiaonan Chen conceived and designed the experiments, performed the experiments, authored or reviewed drafts of the paper, and approved the final draft.

## Data Availability

There is no raw data for this literature review.

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
