# Peer review of "Role of methyltransferase-like enzyme 3 and methyltransferase-like enzyme 14 in urological cancers"

_PeerJ, doi:10.7717/peerj.9589_

## Round 0.1 · original submission · Major Revisions

All critiques of the reviewers should be addressed and the manuscript amended accordingly.

Reviewer 1 ·

Basic reporting

Needs improvement of English.

Experimental design

Needs more recent references to relevant works, specifically on the same topic.

Validity of the findings

Figures and table are helpful for the reader.

Additional comments

In this review the authors summarize evidence on m6A-related proteins METTL3 and METTL14 on urological cancers. I have some suggestions for improving the manuscript:

-Some gramatical errors and sentences merit review: eg. “When METTL3 is high expression”, please rephrase. Sentence “These results suggest that although METTL3 and METTL14 show different expression and regulation mechanism in urological cancers” makes no sense, it is not complete. The same with “Similar to METTL3, METTL14 are more predisposed to mutation or CNV” and “Low METTL14 and PTEN expression show shorter OS” – the patients show mutations or poorer survival, not the proteins, please rephrase. Please change “style” for “type” when referring to the histological types of the various cancers. Prostate cancer is abbreviated, but then the abbreviation is not always used after the first mention. I advise authors to carefully review the written English for these details.

-The authors lack reference to some recent reviews. Including works on the exact same topic, i.e., m6A modifications on Urological Tumors, including METTL3 and METTL14. I suggest adding “The Emerging Role of Epitranscriptomics in Cancer: Focus on Urological Tumors, PMID: 30428628” to the discussion.

-The authors decide to focus on urological cancer, but leave out testicular tumors, which is unfortunate given the insightful parallelism between the impact of m6A during development and the emergence of germ cell tumors, which would be interesting for the reader. I suggest adding this briefly to the review, namely discussing very recent works on this topic, which can substantiate the impact of m6A in urological cancer: “m6A RNA modification and its writer/reader VIRMA/YTHDF3 in testicular germ cell tumors: a role in seminoma phenotype maintenance, PMID: 30866959” and also “N6-Methyladenosine detected in RNA of testicular germ cell tumors is controlled by METTL3, ALKBH5, YTHDC1/F1/F2, and HNRNPC as writers, erasers, and readers, PMID: 30903744”.

Authors are also missing a recent complete work on prostate cancer (“VIRMA-Dependent N6-Methyladenosine Modifications Regulate the Expression of Long Non-Coding RNAs CCAT1 and CCAT2 in Prostate Cancer, PMID 32218194”. Since the authors present a review article specifically directed to urological cancer in a recent expanding field, I advise authors to be complete in their literature search and include/discuss all works available on m6A-related proteins, to discuss briefly and give the reader a broad overview, and only then give focus to METTL3/METTL14.

Reviewer 2 ·

Basic reporting

The manuscript,“Role of methyltransferase-like enzyme 3 and methyltransferase-like enzyme 14 of N6- methyladenosine in the urological cancers” submitted by Zijia Tao and coworkers, describes METTL3 and METTL14 expression in urological cancers. They found a potential possibility for using m6A as a new therapeutic target in urological cancer.

Experimental design

No comment

Validity of the findings

Conclusions are well stated.

Additional comments

This manuscript was well-written and presented in a clear and logical fashion.

Reviewer 3 ·

Basic reporting

The authors have done a good job in assimilating the literature knowledge into this review. However, it might be important for all readers to know what is urological cancer and its types in the "Introduction", which is not well-written. Please include how this review is going to advance the field in the form of significance of this review.

There are few grammatical English language error.

Experimental design

The authors managed to categorize well but this has led to redundancy of many points throughout with proper references.

Validity of the findings

Need to add significance of this review in the existing field.

Additional comments

1. Line 52 “living entities” can be replaced with “living organisms”
2. Line 149-150: meaning is not clear
3. Line 151-157: This sentence is very generalized for all cancers, might not be needed here
4. The authors could use the appropriate inhibitory pathway arrow instead of the yellow arrow.

---

## Round 0.2 · Minor Revisions

Please address the remaining issues pointed out by the reviewers and revise your manuscript accordingly.

Reviewer 1 ·

Basic reporting

No further comments.

Experimental design

No further comments.

Validity of the findings

No further comments.

Additional comments

No further comments.

Reviewer 3 ·

Basic reporting

The authors have thoroughly worked and improved upon the previous remarks made. However, there are few points below which I think will help to make the work more accurate.
1. English language throughout the text is very convoluted with too many jargons and gene symbols, that are explained nowhere, which off-puts readers and leads to confusion than convincing the stated facts. Language clearity will enable the authors to keep the review be well cited.
2. Fig3, I guess the authors wanted to write “less reports of study with METTL14 in prostate cancer” not METTL3
3. Please be consistent with the terminologies like ONCOGENE and TUMOR SUPPRESSOR which should be uniform in text and figures and tables. Avoid writing tumor promoting and anti-oncogenic, they are incorrect.

Experimental design

It is well investigated study, however, the authors should also included TCGA urological cancer datasets.

Validity of the findings

Please check the figure legend of Fig 3.

---

## Round 0.3 · accepted · Accept

Thank you for addressing remaining critiques of the reviewer. I am pleased to let you know that the revised version is acceptable now.